# Effect of Etanercept on *Plasmodium yoelii* MDR-Induced Liver Lipid Infiltration

Bhavana Singh Chauhan [1,2], Sarika Gunjan [1,2], Sunil Kumar Singh [1], Swaroop Kumar Pandey [1,3,*] and Renu Tripathi [1,2,*]

1    Division of Parasitology, CSIR-Central Drug Research Institute, Lucknow 226031, India
2    Academy of Scientific and Innovative Research, New Delhi 110025, India
3    Department of Life Sciences, Ben-Gurion University of the Negev, Beer-Sheva 84105, Israel
*    Correspondence: pandey@post.bgu.ac.il (S.K.P.); renu_tripathi@cdri.res.in (R.T.)

**Abstract:** The lipid is a vital metabolic and structural component of the malaria parasite. Malaria parasite-induced liver lipid deposits undergo peroxidation, which ultimately causes tissue damage and histopathological changes, which further lead to many complications. Therefore, it is essential to focus on the factors responsible for this stimulated lipid accumulation during malaria infection. In the present study, we have correlated the significant increase in serum TNF-$\alpha$ and liver triglyceride during *Plasmodium yoelii* MDR infection in mice. In order to explore the role of TNF-$\alpha$ in inducing lipid accumulation in the liver during malaria infection, we have used a competitive TNF-$\alpha$ inhibitor Etanercept, for the treatment of *Plasmodium yoelii* MDR (*Py* MDR) infected mice and found that Etanercept displayed up to a three-fold inhibition of the liver triglyceride level in *Py* MDR infected mice. These results were also confirmed by triglyceride specific oil red O staining of liver sections. In addition, all the treatment groups also showed inhibition in the level of serum TNF-$\alpha$ and the liver malondialdehyde (MDA), a byproduct of lipid peroxidation. Our study thus concludes that Etanercept significantly reduces Plasmodium-induced liver triglyceride and further saves the host liver from malaria-induced lipid infiltration and liver damage. Therefore, treatment with Etanercept, along with a standard antimalarial, may prove a better therapy for the disease.

**Keywords:** etanercept; TNF-$\alpha$; triglyceride; malondialdehyde (MDA); malaria; MDR; liver pathology

## 1. Introduction

Malaria, caused by the intracellular protozoan parasite of the genus Plasmodium, is one of the world's major infectious diseases. There were an estimated 241 million malaria cases and 627,000 malaria deaths worldwide in 2020 [1]. Prevention of malaria relies mostly on chemotherapy (antimalarials especially artemisinin and its derivatives). A few years back, there were reports of artemisinin resistance from the Greater Mekong subregion [2] and Equatorial Guinea, and the threat of its spread to other areas is an emerging problem [3]. Vaccination is considered another important control measure, which has shown slow progress. Recently, a malaria vaccine candidate, RTS, S, has shown new hope [1], however, it has some economic and community health issues [4]. Therefore, to continue our fight against malaria, it is a prerequisite to identify new drug targets and vaccine candidates. For the determination of new targets, the mechanism of survival and the pathogenesis of Plasmodium should be deeply explored. Pathogenesis induced by the parasite led to an adverse effect on the host body. One of the pathologies caused by Plasmodium during infection is lipid deposition in the liver and kidney of the host displayed by Sudan black B staining [5] and this lipid undergoes peroxidation [6,7] that leads to liver damage. A lipid is also vital for parasite survival due to its essentiality in β-hematin formation, characterized as a neutral lipid blend (NLB), and monopalmitoylglycerol (MPG). This process occurs inside the food vacuole (FV) of the parasite [8]. Therefore, factors inducing lipid deposition in a host at the time of infection should be explored intensely.

The pathogenesis of malaria is complex and includes both immunologic and nonimmunologic aspects. In the immunological aspect, tumour necrosis factor alpha (TNF-$\alpha$) has a key role. Previous reports indicated that TNF-$\alpha$ has the ability to induce pathogenesis in uncomplicated, as well as complicated, malaria. TNF-$\alpha$ stimulates the endothelial cells of the brain to express ICAM-1, and vessels in the brain have further amplified the expression of ICAM-1 during cerebral malaria [9,10]. TNF-$\alpha$, viewed as a potential pathogenic factor, can contribute either directly or indirectly to many pathological processes, but the role of TNF-$\alpha$ in induced lipid deposition during Plasmodium infection has not been explored yet.

TNF-$\alpha$ is a proinflammatory cytokine and is produced mainly by activated monocytes/macrophages [11]. It is synthesized as a transmembrane protein (tmTNF) cleaved by the matrix metalloproteinase (TACE) to soluble TNF [12]. TNF-$\alpha$ acts through its receptors, and these are TNFR1 (p55) and inducible TNFR2 (p75) [13]. Both these receptors are membrane glycoproteins, but they vary in expression, ligand affinity, and signaling pathway activation. Various TNF inhibitors are present in the market to restrict the pathology induced by TNF-$\alpha$. One of them, Etanercept (Enbrel), an anti-TNF-$\alpha$ agent, is a fusion protein of the human p75 TNF-$\alpha$ receptor attached to the Fc portion of human IgG1 [14], which the FDA has approved for the treatment of rheumatoid arthritis [15]. However, no studies are conducted on the effect of Etanercept against TNF-induced lipid pathology during Plasmodium infection.

To study the effect of Etanercept on malaria, we designed the present study, where we evaluated the effect of its treatment on liver lipid level, a marker of oxidative stress and parasite profile during infection.

## 2. Materials and Methods

### 2.1. Experimental Model

Laboratory-bred Balb/c mice of either sex, weighing 20–22 g, were obtained from breeding colonies of the National Laboratory Animal Centre at CSIR-CDRI, Lucknow, and were used for the experiments. Mice were kept in polypropylene cages in groups of 5–6 animals/cage at the Institute's animal facility under certified conditions of temperature (22–26 °C) and relative humidity (60–70%), with free access to food and water. Experimental studies were approved by the 'Institutional Animal Ethics Committee' recognized by CPCSEA, Ministry of Environment & Forests, Government of India (IAEC/2008/117/Renew 09(49/16)). All procedures performed in studies involving animals were in accordance with the ethical standards of the institution or practice at which the studies were conducted.

### 2.2. Treatment with Etanercept

Etanercept (Cipla, Mumbai, India) was prepared in phosphate-buffered saline (PBS) and three different doses were used for subcutaneous inoculation i.e., $0.5 \times 1$, $1 \times 1$, and 5 mg/kg $\times 1$. Treatment was prophylactic (a day before infection) followed by inoculation of $1 \times 10^5$ *Py* MDR infected RBCs/mouse. Parasitaemia was monitored daily post infection and determined by counting at least 1000 erythrocytes from thin blood smears stained with Giemsa. When parasitaemia reached up to 30–45%, the mice were sacrificed, blood was collected for serum isolation, and the liver was excised for further histopathological, histochemical, and biochemical studies.

### 2.3. Cryosectioning of Tissue

Fixed tissues were transferred to 20% sucrose solution a day before cryosectioning (cryostat: LEICA CM1850, Leica, Wetzlar, Germany). After overnight incubation at 4 °C, tissues were processed for cryosectioning (sectioning at −20 °C). Blocks were prepared by using OCT (10% polyvinyl alcohol and 4% polyethylene glycol), kept at −20 °C (until ready for sectioning) and then the blocks were sectioned and taken on the slide. Slides were processed for staining. Slides were stored at −20 °C until staining. Stained slides were examined under a light microscope (Nikon Eclipse E200, Nikon, Tokyo, Japan). Staining of liver sections was conducted with lipid-specific stains. Sudan black [5] was used for total lipid and oil red O staining [16] was neutral lipid specific.

### 2.4. Microtomy and Hematoxylin-Eosin (HE) Staining

Tissues were fixed in 10% neutral buffered formalin for 48 h at 4 °C and then processed for paraffin block preparation. The paraffin block containing liver tissue was first trimmed at 15 microns to remove excess wax and sectioned into 5 microns sections. The sections were then deparaffinised and stained with hematoxylin and eosin (H&E). Slides were examined under a light microscope (Nikon Eclipse E200, Nikon, Tokyo, Japan).

### 2.5. Lipid Quantification

Lipid concentration in the liver was quantified by using a commercially available kit (Cayman TG kit) according to the manufacturer's instructions. Briefly, 350–400 mg of liver tissue was homogenized in 2 mL of the diluted standard diluents and centrifuged at 10,000 g for 10 min at 4 °C. Before assaying samples, the supernatant was diluted 5 fold. For the assay, 10 μL of the serially diluted standard and 10 μL of the sample were added onto the plate. The reaction was initiated by adding 150 μL of the diluted enzyme buffer solution to each well. Then the plate was incubated for 15 min at room temperature. Absorbance was recorded at 530–550 nm. The value of triglyceride was calculated using the linear regression of the standard curve.

### 2.6. Lipid Peroxidation Assay

A lipid peroxidation assay was performed as previously described [17]. Briefly, the liver was homogenised in 1.5% KCl, 10% liver homogenate was prepared and incubated for 10 min at 37 °C. One ml of the homogenate was mixed with 1.5 mL of 10% trichloroacetic acid (Sigma-Aldrich, St. Louis, MA, USA), and centrifuged for 10 min at 2000 rpm. The supernatant was collected and two ml of supernatant was mixed with 2 mL of 0.67% thiobarbituric acid (Sigma-Aldrich), heated for 15 min in a boiling water bath, cooled, centrifuged at 1000 g for 10 min, and supernatant volume adjusted up to 5 mL with MilliQ water. The Absorption of the colour reaction was measured in a spectrophotometer at 535 nm against a blank prepared with one ml of phosphate-buffered potassium chloride in place of the homogenate. The absorbance at 535 nm was expressed as μ mol of malonyl-dialdehyde per gm wet tissue.

### 2.7. ELISA for Determination of Serum TNF-α

The concentration of TNF-α was determined in serum obtained from coagulated blood (15 min at 37 °C then 30 min at 4 °C, stored at −20 °C until analysis) by ELISA using commercially available kits (BD OptEIA™; BD Biosciences, Franklin Lakes, NJ, USA) according to the manufacturer's instructions. Briefly, capture antibody was coated a day before and plates were left at 4 °C. Next day, after 1 h of blocking in PBS with 10% FBS, samples were incubated for 2 h, detection antibody was added for 1 h, streptavidin-avidin incubated for 1 h followed by the addition of 3,3,5,5 tetramethylbenzidine (TMB) substrate and absorbance was measured at 450 nm.

### 2.8. Statistical Analysis

Statistical significance between different groups was determined using Student's *t* test with significance set at the levels of $p < 0.05$, $p < 0.001$ and $p < 0.0001$, which corresponds to significant, highly significant and most significant values respectively. All the data are presented as mean $\pm$ SEM values. All statistical analyses were performed with Graph Pad Prism 5.0 Software.

## 3. Results

This section may be divided by subheadings. It should provide a concise and precise description of the experimental results, their interpretation, as well as the experimental conclusions that can be drawn.

### 3.1. Parasitaemia Profile after Etanercept Treatment

The percent of parasitaemia of all the groups was recorded after infection with Py MDR, and on day four it was $0.84 \pm 0.56$, $0.94 \pm 0.43$, $0.80 \pm 0.57$, and $1.33 \pm 0.65\%$ after prophylactic treatment of 0.5, 1, 5 mg/kg doses of Etanercept and infected control, respectively. As expected, parasitaemia inhibition was not prominent, and it was approximately comparable to infected control. Parasite inhibition was 20, 35, and 35% respective to infected control in 0.5, 1 and 5 mg/kg treatment groups respectively (Figure 1).

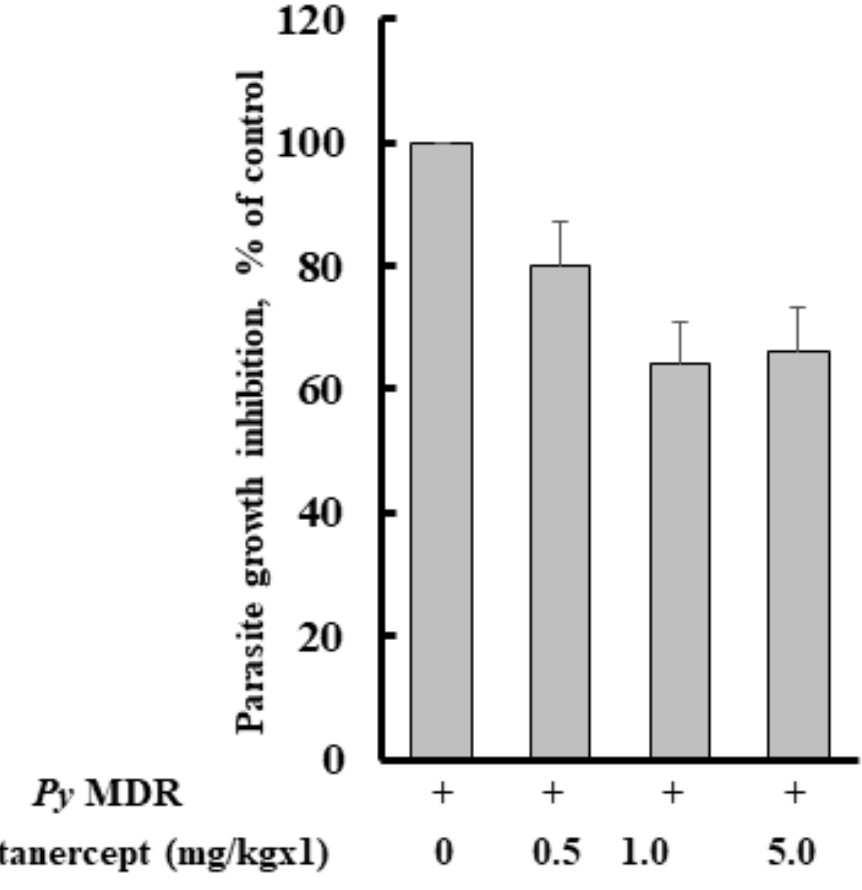

**Figure 1.** Effect of Etanercept treatment on *Py* MDR infection in Swiss mice. Etanercept was prepared in PBS and given subcutaneously just after intraperitoneal inoculation of $1 \times 10^5$ *Py* MDR infected RBCs/mouse (*n* = 6). Parasitaemia was monitored on day four post infection and determined by counting at least 1000 erythrocytes from thin blood smears stained with Giemsa under a $100\times$ microscope.

### 3.2. TNF-α and Liver Triglyceride Level Increases during Py MDR Infection

To establish the correlation between serum TNF-α level and liver triglyceride in malaria infection, we assessed TNF-α in infected and healthy (uninfected) mice livers and blood samples. Quantification of triglycerides in the liver of *Py* MDR infected mice at a 30–45% parasitaemia range showed a significant increase in its content as compared to uninfected control. Triglyceride content was also increased up to six fold in an infected liver as compared to an uninfected liver ($p$ *** $< 0.0001$). The concentration of triglyceride in an uninfected mice liver was found to be $30 \pm 3.3$ mg/dL, while it was $180 \pm 5.9$ mg/dL in infected mice liver at the same parasitaemia range. Serum triglyceride in an uninfected mice liver was also significantly higher than in uninfected mice (1.5 fold), as compared to the uninfected group (Figure 2A). We determined the changes in serum TNF-α during infection and found that the level of serum TNF-α was increased significantly ($p$ ** $= 0.001$), with the mean concentration of $143.8 \pm 23.0$ pg/mL in the infected mice group, whereas it was recorded at very a low concentration in uninfected mouse serum, i.e., $22.8 \pm 3.3$ pg/mL (Figure 2B).

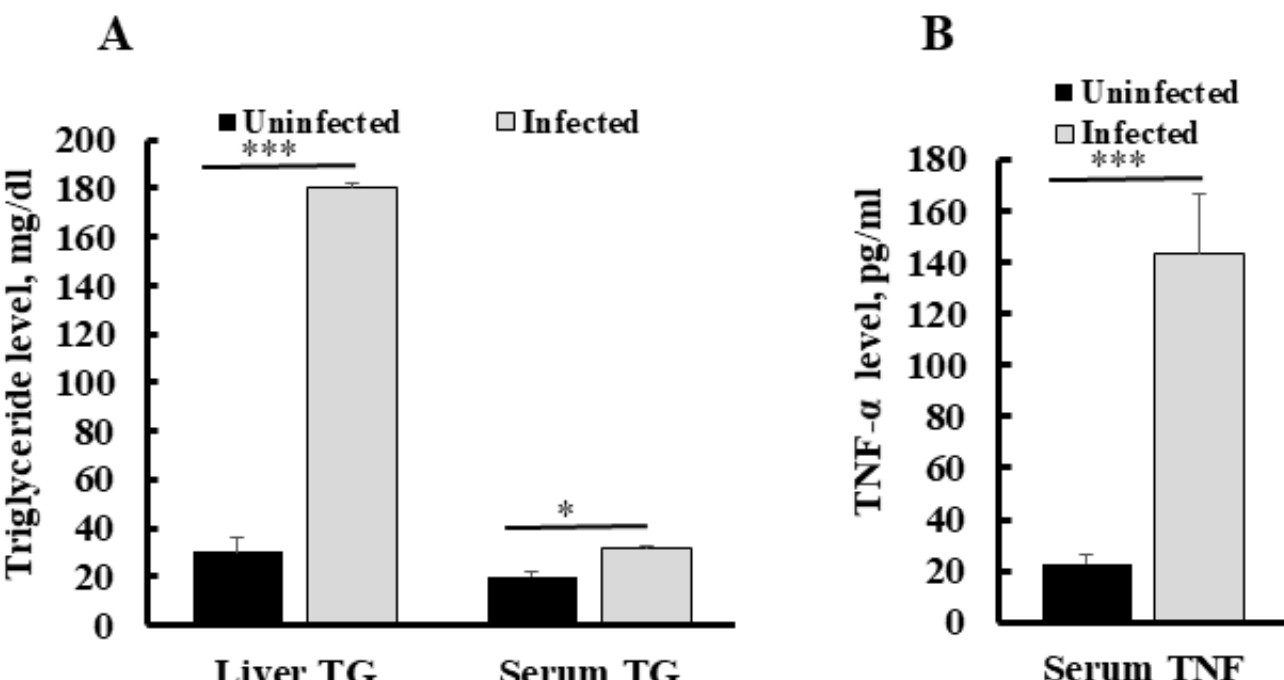

**Figure 2.** Concentration of serum TNF-α and liver triglyceride during *Py* MDR infection as compared to uninfected control (**A**) A significant increase in serum TNF-α level was observed in *Py* MDR infected mice, as compared to uninfected controls. (**B**) A significant increase in liver triglyceride was observed when *Py* MDR infected mice livers were compared with uninfected controls. $p$ * $< 0.05$, $p$ *** $< 0.0001$.

### 3.3. Etanercept Treatment Reduces Liver Triglyceride, MDA (Malondialdehyde), and Serum TNF-α in Py MDR Infected Mice

When the parasitaemia reached up to 30–40% in infected mice, the level of triglyceride in the liver was significantly elevated to $182 \pm 2.4$ mg/dL, as compared to uninfected/healthy control ($28 \pm 4$ mg/dL) ($p$ *** $< 0.0001$), a six-fold increase was recorded. Results of prophylactic Etanercept treatment demonstrated a significant decrease in liver TG when compared to infected control ($p$ *** $< 0.0001$). Significant declines of 2.8, 2.8 and 2.4 folds were recorded in the 0.5, 1 and 5 mg/kg treatment group respectively. The concentration of liver TG was $63 \pm 0.6$, $65 \pm 4.9$ and $73 \pm 3.4$ mg/dL after 0.5, 1 and 5 mg/kg doses of Etanercept, respectively (Figure 3A).

Quantitative studies demonstrated that prophylactic treatment with different doses of Etanercept inhibited the parasite induced triglyceride level in the treated liver as compared to infected liver. This study was confirmed by oil red O (neutral lipid) staining of the liver cryosections (Figure 3B). Staining of cryosections from the 5 mg/kg treated group showed the reduced triglyceride level in the liver as compared to infected control. Deposition of lipid droplets in hepatocytes was reduced. The livers of healthy mice showed scanty deposition of triglyceride as compared to infected control, which displayed massive deposition of the same throughout the section. A similar pattern was seen with Sudan black staining (total lipid); it also demonstrated the same results (Figure 3C). Etanercept also reduced the level of total lipid in different treatment groups; total lipid had shown the same reduction pattern as oil red O. Liver sections of uninfected control showed a proper network of hepatocytes around the central vein and constricted sinusoids, while infected control showed a widened sinusoid, hemozoin pigment deposition, and a distorted network of hepatocytes. Etanercept-treated mice liver displayed morphology similar to uninfected control (Figure S1).

The level of malondialdehyde, a byproduct of lipid peroxidation, was increased significantly up to five fold in the infected liver as compared to uninfected control. In treatment groups, a significant decrease in liver MDA was observed. Inhibition was about 2, 2.5 and 2.5 fold in 0.5, 1 and 5 mg/kg treatment groups respectively (Figure 4A). All the treatment groups showed a similar decline in MDA level when compared to infected control. Results indicated that treatment was able to inhibit parasite induced lipid peroxidation.

We determined and compared the TNF-α level after prophylactic treatment of etanercept in malaria infected mice with uninfected control mice. The TNF-α level was significantly reduced in all the treatment groups as compared to infected control. (Figure 4B).

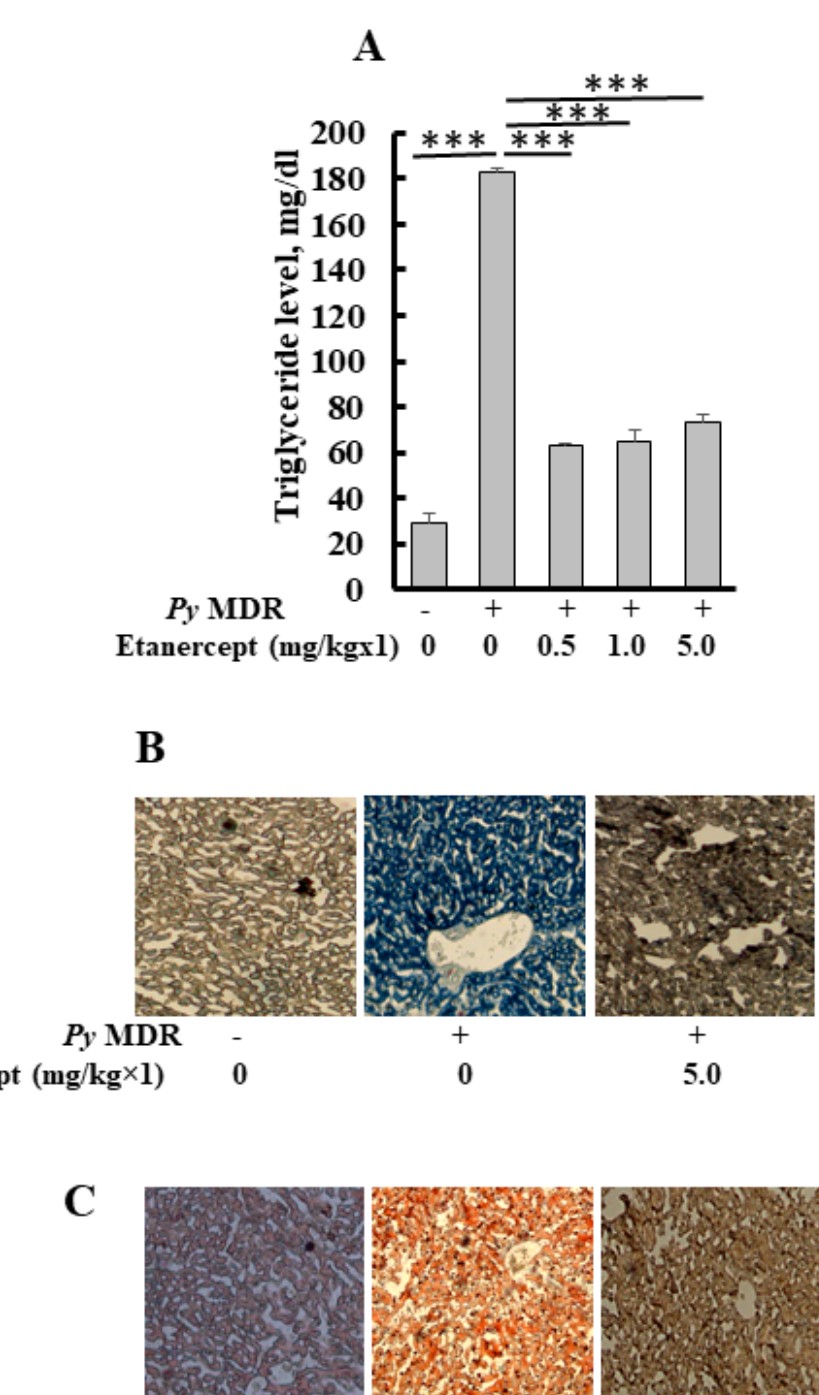

**Figure 3.** Quantification of triglyceride, oil red O and Sudan black B staining of liver sections. (**A**) Level of triglyceride in the liver after prophylactic treatment of Etanercept (**B**,**C**). Images show representative oil red O stained (**B**) Sudan black B (**C**) liver section from uninfected *P. yoelii* MDR-infected and Etanercept, 5 mg/kg×1 treated mice groups. ($p$ *** < 0.0001). During malaria infection, histological changes in the liver of infected mice were similar to the dilatation of sinusoids, a distorted network of hepatocytes, and deposition of hemozoin. Etanercept was administered subcutaneously, followed by infection to prove the role of TNF-α in stimulated lipid deposition during infection, which leads to tissue damage. Our previous results indicated that the inhibitor was capable of reducing the lipid deposition.

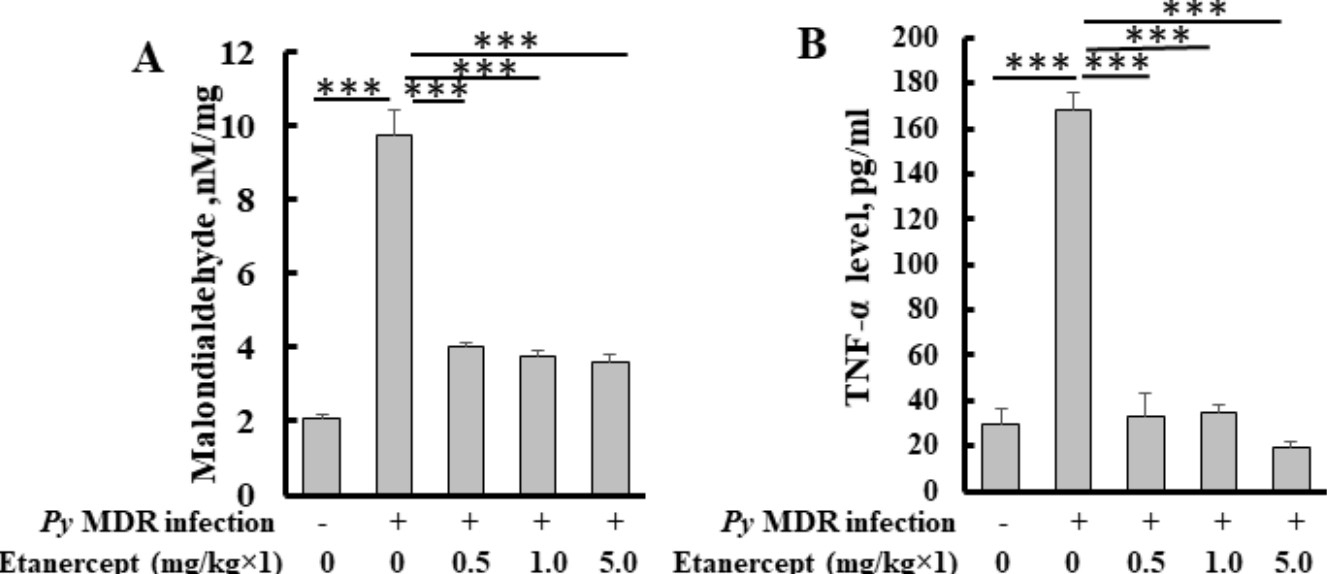

**Figure 4.** Quantification of MDA (**A**) and TNF-α (**B**) level in the livers of Etanercept treated mice. *** *p* < 0.0001.

## 4. Discussion

The current study extends our previous findings, wherein, we had reported an increase in hepatic lipid deposits in the liver during malaria infection [5], to the identification of factors responsible for this stimulation. Earlier studies have demonstrated that administration of TNF-α stimulates lipid (plasma TG, cholesterol) and sterol synthesis in rat liver, which can be seen 2 h after TNF-α infusion and lasted for 17 h. The proinflammatory cytokine, TNF-α, is a soluble mediator, and released after Plasmodium infection [18] and its plasma level correlates with the severity of malaria disease [19]. Besides, overproduction of TNF-α is a fundamental mechanism of autoimmune diseases, including RA (rheumatoid arthritis), ankylosing spondylitis, and psoriatic arthritis [20].

The role of TNF-α in lipid accumulation and related pathology during malaria infection has not been explored yet. Hence, this is the first report, wherein we have shown the correlation between an increase in serum TNF-α during *Plasmodium yoelii* infection and an elevation in the liver lipid deposits. We observed an abundant presence of neutral lipid in the infected mouse liver, and we suggest that this intensification is due to overproduction of TNF-α during infection. We performed a study to test the hypothesis that TNF-α has an essential role in elevated lipid deposits during malaria infection, which would make it a promising target for future therapies. In addition, we have shown that Etanercept (Enbrel), a widely used TNF-α antagonist, attenuates malaria induced liver neutral lipid deposition and tissue damage. Earlier blocking of excess TNF-α by its antagonists, including TNF receptor 2-IgG1 fusion protein (Etanercept), had been validated as an effective treatment for RA [21,22].

For confirmation of the role of TNF-α in Plasmodium induced liver lipid deposition during infection, groups of mice were prophylactically (before infection) treated with its inhibitor, i.e., Etanercept, at three different doses, 0.5, 1, and 5 mg/kg. Etanercept comprises the extracellular region of human TNFR2 expressed as a fusion protein with a C-terminal human IgG1 crystallizable fragment (Fc) domain [23]. Although, TNF-α is an immunomodulator, still it exhibited a low level of suppression as evident in 0.5, 1, and 5 mg/kg treatment groups. A complete suppression of TNF-α is not recommended as it has a protective inflammatory role in the host during early malaria infection. Total inhibition of this cytokine led to a rapid increase in parasitaemia. A case was reported with overwhelming parasitaemia in a patient undergoing treatment with infliximab for rheumatoid arthritis [24]. Our results demonstrated slow progression of parasitaemia and

little inhibition after treatment with a TNF-$\alpha$ inhibitor. This shows that doses of the TNF-$\alpha$ inhibitor used in our experiment did not amplify the parasitaemia, as described previously. The present study suggests that for the improved decline or total inhibition of parasitaemia, Etanercept may be used in combination with existing antimalarials.

Blocking TNF-$\alpha$ obviously inhibited the overproduction of this inflammatory cytokine as, after treatment of mice with the inhibitor, the level of serum TNF-$\alpha$ showed a significant decline of about five to nine fold in its concentration, as compared to infected control. Our study indicates that the Etanercept treatment inhibits the Plasmodium induced serum TNF-$\alpha$ concentration.

Further, we showed that blocking TNF-$\alpha$ by using Etanercept inhibited the liver triglyceride level to about three fold, even with the lowest dose of Etanercept. Hence, this brings us closer to our hypothesis by showing that Etanercept, an agent that is already in wide clinical use for other indications, is effective in reducing liver lipid deposits. These results are similar to the previous reports in which treatment with the TNF-$\alpha$ blocker, i.e., Etanercept, to a group of children with JIA (juvenile idiopathic arthritis) showed that TC (total cholesterol), LDL (low-density lipoprotein) cholesterol, and TG (triglyceride) were significantly reduced after six and twelve months of therapy [25]. Quantitative results were confirmed by the staining of liver sections with oil red O and Sudan black B that stained triglyceride and total lipid, respectively, and demonstrated that triglyceride and total lipid were inhibited in treatment groups. There was a sporadic deposition of lipids in liver sections of treatment groups, as compared to massive deposition of lipids in the infected liver section. Our results confirm that TNF-$\alpha$ triggers the liver triglyceride during Plasmodium infection, and so blocking TNF-$\alpha$ inhibits the accumulation of triglyceride in the liver.

Lipids could probably react with oxygen radicals and undergo peroxidation [26]. Earlier reports have also shown that Py nigeriensis infection results in a significant increase in hepatic oxidative stress indices viz., rate of lipid peroxidation (LPO) [6]. Among the many diverse aldehydes, which can be formed as secondary products during lipid peroxidation, malondialdehyde (MDA) and other byproducts have been extensively studied [27–29]. Prior reports have revealed increased levels of lipid peroxidation markers, such as MDA, which have been implicated in a number of diseases, including malaria [30,31]. In the present study, we attempted to determine whether Plasmodium-induced liver MDA level was affected by Etanercept, and about a three-fold inhibition in the liver MDA level was observed in 0.5, 1, and 5 mg/kg treatment groups. A significant decline in liver MDA indicates a reduction in lipid peroxidation, hence it shows that TNF-$\alpha$ has an essential role in inducing lipid peroxidation during malaria infection. A study conducted on rheumatoid arthritis patients showed that Etanercept acts as a regulator against oxidative DNA damage, and lipid peroxidation [32]. MDA appears to be the most mutagenic product of lipid peroxidation; it is a potentially vital contributor to DNA damage and mutation that is produced endogenously via lipid peroxidation [29]. This further strengthens the evidence that TNF-$\alpha$ inhibition is beneficial regarding inhibition of hepatic hypertriglyceridemia and lipid peroxidation.

Induced lipid peroxidation during infection leads to liver damage as displayed by histopathological changes in the liver sections. The previous reports had documented hyperplasic Kupffer cells with scattered haemozoin deposition, portal inflammation and sinusoidal infiltration and dilatation as the most common histopathological changes in the livers of severe *P. falciparum* malaria cases [33]. Importantly, the anti-TNF$\alpha$ treatment improved the histopathological changes that arose during malaria infection. We observed that only a 5 mg/kg dose of Etanercept approximately restored normal histology, hepatocytes were intact, and sinusoids were narrowed. Anti-TNF therapy has shown its explicit efficacy against hepatic damage caused by Plasmodium infection in the rodent model.

## 5. Conclusions

TNF-α is a proinflammatory cytokine, and during malaria infection it has both inflammatory as well as pathogenic roles which adversely affect the host. Our results suggest that elevation of TNF-α plays an important role in stimulating profuse liver triglyceride and liver MDA level that are the hallmark features of lipid peroxidation, ultimately cause liver damage. The dropping of TNF-α levels by using Etanercept is effective in reducing the progression of neutral lipid deposition, liver MDA, and tissue damage. It protects the host from malaria-induced lipid pathology. The present study brings us closer to clinical translation by showing that Etanercept, an agent that is already in wide clinical use for other indications, is effective and may be explored in combination with effective antimalarials for the treatment of severe malaria-induced pathology.

**Supplementary Materials:** The following supporting information can be downloaded at: https://www.mdpi.com/article/10.3390/futurepharmacol2040031/s1, Figure S1: HE stained liver section after treatment with etanercept. Liver section of uninfected control showed proper network of hepatocytes around central vein and constricted sinusoids, while infected control showed widen sinusoid, hemozoin pigment deposition, and distorted network of hepatocytes. Etanercept treated mice liver displayed morphology similar uninfected control.

**Author Contributions:** Conceptualization, R.T. and S.K.P.; methodology, B.S.C., S.G. and S.K.S.; validation, B.S.C., S.K.P. and R.T.; formal analysis, B.S.C.; S.G. and S.K.S.; investigation, R.T. and S.K.P.; resources, R.T.; data curation, B.S.C.; writing—B.S.C.; writing—review and editing, R.T. and S.K.P.; visualization, B.S.C.; supervision, R.T.; project administration, R.T.; funding acquisition, R.T. All authors have read and agreed to the published version of the manuscript.

**Funding:** This research was funded by SPlenDID-CSIR India to R.T "Emerging and re-emerging challenges in infectious diseases: Systems based drug design for infectious diseases".

**Institutional Review Board Statement:** The study was conducted in accordance with the 'Institutional Animal Ethics Committee' recognized by CPCSEA, Ministry of Environment & Forests, Government of India (IAEC/2008/117/Renew 09(49/16). CDRI MS. no. 10487.

**Informed Consent Statement:** Not applicable.

**Data Availability Statement:** Not applicable.

**Acknowledgments:** We are thankful to the Council of Scientific and Industrial Research Centre (CSIR), India, for funding to perform this study. The authors also thank the Director Central Drug Research Institute (CDRI), for extending all the necessary facilities. We are also thankful to Sheeba Thomas for helping in the cryosectioning of tissues.

**Conflicts of Interest:** The authors declare no conflict of interest.

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
