# Peer review of "Effect of Etanercept on Plasmodium yoelii MDR-Induced Liver Lipid Infiltration"

_futurepharmacol, doi:10.3390/futurepharmacol2040031_

Round 1

Reviewer 1 Report

The study is relevant showing that TNF-α plays an important role in stimulating profuse liver triglyceride and liver MDA level that is the hallmark features of lipid peroxidation, ultimately causing liver damage. This is the first study indicating that Etanercept is effective in dropping levels of TNF-α during malaria. Etanercept has been previously approved for the treatment of rheumatoid arthritis. Authors suggest that treatment with Etanercept along with standard antimalarial may prove as a better therapy for the disease.

Species name Plasmodium yoelii have to be in italic throughout manuscript

From the study is not clear what is the impact of standard antimalaria drugs to the level of analysed compounds such as MDA, tryglicerides, TNF-α.

L81-82 please provide number of approvals for the experimental study with animals

L100-101 it is not clear, for how long block were kept at -20C or promptly analyzed

L121 reference should be in square brackets

L124,125, 136 country of “Sigma” should be indicated, isn’t it Sigma-Aldrich?

L101 please include, country of microscope, Japan

Author Response

We are thankful to the learned reviewer for valuable suggestions. We have tried our best to clarify the questions and incorporated the valuable suggestions (attached). A pointwise clarification is provided in the revised manuscript.

Reviewer 2 Report

The authors designed this study to explore the role of TNF-α in inducing lipid accumulation in the liver (and subsequently liver damage) during malaria infection in mice infected with Plasmodium yoelii MDR by using a competitive TNF-α inhibitor etanercept which was administered s.c. The results indicate that etanercept displayed up to 3 folds inhibition of liver triglyceride level in Py MDR infected mice. All the treatment groups also showed inhibition in the level of serum TNF-α and the liver malondialdehyde (MDA), a by-product of lipid peroxidation. I have no special comments, the paper is well written. Authors could run through the manuscript and correct minor typo errors.

Author Response

(The authors gave the same response as above.)

Reviewer 3 Report

The authors of the manuscript: “Effect of Etanercept on Plasmodium yoelii MDR induced liver 2 lipid infiltration” show that Enbrel can significantly reduce the levels of triglycerides in the liver of Plasmodium yoelii-infected mice.

1.    The authors indicated that Enbrel could be used as adjunctive therapy in malaria. Despite lower levels of TG and TNF in P. yoelii-infected mice, there is no evidence that reduction of liver TG is beneficial to mice. Mice are still infected with malaria parasites both in the other organs and even if TG are significantly, parasites can still bind to endothelial cells and cause tissue damage. Would be nice to have measures of better clinical outcomes in infected mice receiving versus mice that are not receiving Enbrel.

2.    The experimental design and approach would benefit from schematic representation of the mice indicating the days pre- and post-infection with all the measurements taken. Having a mouse weight monitoring overtime would be helpful at least as a supplemental information. What is the effect of Enbrel in anemia? Does Enbrel affect host innate and adaptive immunity? Can you save mice with Enbrel?

3.    What were the levels of hepatic enzymes AST and ALT in Enbrel-treated versus Enbrel untreated?

4.    Higher levels of lipids in untreated mice may be associated higher levels of anti-oxidant enzymes such as Catalase and GSH. Have the authors investigated this previously?

5.    What would happen in the liver of malaria-infected mice if treatment was started at the time of latent parasitemia or 1 to two days post-infection?

6.    Two of the most common side effects of Enbrel are chills and fever. Chills and fever are the most common clinical symptoms of malaria. What do the authors think about this?

7.    TNF is needed in malaria. It has a protective role by shaping innate and adaptive immunity. Targeting TNF as an adjunctive therapeutic strategy has not worked in clinical trials.

8.    TG accumulation in the liver is seen in other mouse models. Is there a mouse model of malaria without accumulation of TG? If yes, that would be your best negative control for assessing the clinical benefit of Enbrel in malaria. Alternatively, Plasmodium berghei NK65 infection in mice is associated with high TGs in the mouse liver. Can Enbrel delay the development of experimental cerebral malaria in mice?

9.    Plasmodium yoelii must be italicized throughout the manuscript.

10. Liver histological staining in Figure 3 could have been better. Authors could use higher magnification pictures as well to show structure of tissues in both cases. Alternatively, a clear H&E slide should be included, and tissues could have been assessed by a board-certified veterinary pathologist blinded to the study groups. How were the tissue sections assessed?

11. If mice were treated with 10 mg/kg of Enbrel, what would you expect?

12. What are the limitations of the study?

Author Response

(The authors gave the same response as above.)

Round 2

Reviewer 3 Report

The authors took care of my concerns during the initial review.